# Effect of Laser-Textured Cu Foil with Deep Ablation on Si Anode Performance in Li-Ion Batteries

**DOI:** 10.3390/nano13182534

**Published:** 2023-09-11

**Authors:** Jingbo Wang, Li Cao, Songyuan Li, Jiejie Xu, Rongshi Xiao, Ting Huang

**Affiliations:** High-Power and Ultrafast Laser Manufacturing Lab, Faculty of Materials and Manufacturing, Beijing University of Technology, Beijing 100124, China

**Keywords:** Si anode, Cu current collector, laser texturing, deep ablation, electrochemical performance, hierarchical micro/nanostructure

## Abstract

Si is a highly promising anode material due to its superior theoretical capacity of up to 3579 mAh/g. However, it is worth noting that Si anodes experience significant volume expansion (>300%) during charging and discharging. Due to the weak adhesion between the anode coating and the smooth Cu foil current collector, the volume-expanded Si anode easily peels off, thus damaging anode cycling performance. In the present study, a femtosecond laser with a wavelength of 515 nm is used to texture Cu foils with a hierarchical microstructure and nanostructure. The peeling and cracking phenomenon in the Si anode are successfully reduced, demonstrating that volume expansion is effectively mitigated, which is attributed to the high specific surface area of the nanostructure and the protection of the deep-ablated microgrooves. Moreover, the hierarchical structure reduces interfacial resistance to promote electron transfer. The Si anode achieves improved cycling stability and rate capability, and the influence of structural features on the aforementioned performance is studied. The Si anode on the 20 μm-thick Cu current collector with a groove density of 75% and a depth of 15 μm exhibits a capacity of 1182 mAh/g after 300 cycles at 1 C and shows a high-rate capacity of 684 mAh/g at 3 C.

## 1. Introduction

Li-ion batteries (LIBs) are important energy storage devices in a variety of industries [1,2]. Si is a promising anode material for LIBs, with a high theoretical specific capacity of up to 3579 mAh/g [3,4]. However, Si experiences a significant volume expansion (>300%) during charging and discharging, resulting in rapid degradation of battery capacity [5] and limiting its practical application.

The current research on Si anodes has focused on nano-sized Si materials [6], high-performance electrolytes [7], binders [8], and structured electrodes [9]. The current collector, as the key component that connects the internal and external circuits of the battery, is used to collect the current generated by the active materials and carry the electrode. The commercial electrode is made by uniformly coating a slurry consisting of active material, conductive additive, and binder on the surface of the current collector. The interfacial adhesion between the electrode coating and the current collector is a primary factor determining the electrode’s structural stability. However, the smooth surface of the commercial Cu foil current collector results in a small contact area with the electrode and weak adhesion. This makes it difficult for the Cu foil to withstand the stress and deformation caused by the volume expansion of the Si anode [10]. As a result, the electrode is highly susceptible to cracking or peeling away from the current collector surface during cycling, leading to poor structural integrity and insufficient electronic conduction of the entire electrode [11]. Therefore, the interfacial property between the Cu foil current collector and the Si anode has been widely regarded as one of the most important factors influencing electrochemical performance [12].

Surface texturing has the potential to enhance the interfacial adhesion between the electrode coating and the current collector [5,13]. Among various surface texturing processes, laser texturing stands out for its flexibility and efficiency, making it easily integrated into industrial assembly lines [14]. In recent years, femtosecond lasers have gained significant attention for texturing current collectors due to their low thermal damage and high accuracy [15]. Most studies have focused on improving conventional graphite anodes. For example, Wang et al. [16] develop a hierarchical micro/nanostructure on a commercial 12 μm-thick Cu foil current collector, resulting in a surface roughness of 485 nm. When applied to a graphite anode, the textured Cu foil demonstrates a capacity retention of 74.7% after 200 cycles. In another study, Zhang et al. [17] create a 500 nm-deep periodic circular pattern on the surface of a 10 μm-thick Cu foil. Compared to the original electrode, the retained capacity increases by 30% after 100 cycles. There has been limited research on enhancing the interfacial adhesion of Si anodes through current collector texturing. Previous studies have primarily utilized surface structures with shallowly ablated grooves (typically on the nanometer scale) or blind holes, which pose challenges for the easy penetration of the electrode slurry. This limitation may restrict the effective mitigation of the substantial volume expansion associated with Si anodes [18].

In summary, while significant progress has been made in texturing current collectors for graphite anodes, there remains a dearth of research focusing on enhancing the interfacial adhesion of Si anodes. Previous studies have utilized surface structures that may not adequately address the volume expansion challenges associated with Si anodes. In this study, we employ a femtosecond laser with a wavelength of 515 nm to texture Cu foil current collectors with different thicknesses (4 μm, 9 μm, and 20 μm) for mitigating volume expansion and improving the interfacial adhesion of the Si anode. On the 4 μm-thick Cu foil, only nanostructures form on the surface, whereas the 9 μm-thick and 20 μm-thick Cu foils exhibit deep microgrooves, with groove depths exceeding 50% of Cu foil thickness. We conduct a comprehensive investigation into the influence of the textured surface on the electrochemical performance to gain insight into how surface texturing of the current collector can improve the cycle stability and rate performance of the Si anode.

## 2. Materials and Methods

### 2.1. Laser Texturing Process

Surface texturing of Cu foils with thicknesses of 4 μm, 9 μm and 20 μm was performed using a femtosecond laser (TRUMPF TruMicro 5280, Germany). The laser parameters were as follows: wavelength of 515 nm, pulse width of 800 fs, and beam diameter of 50 μm. A galvanometer (SCANLAB HurrySCAN14, Germany) controlled the laser beam scanning at a scanning rate of 1000 mm/s. The laser texturing was carried out at a repetition rate of 100 kHz and a laser fluence of 3.18 J/cm^2^.

### 2.2. Characterization

Surface morphology and composition of the Cu foils were examined using a scanning electron microscope (SEM, HITACHI S-3400N, Japan) equipped with an energy-dispersive spectrometer (EDS, Bruker XFlash 5010, Germany). The ablation depth was measured using a laser scanning confocal microscope (LSCM, KEYENCE VHX-950F, China). The wettability of the samples was evaluated by measuring the static contact angle using a video-based contact angle meter (BIOLIN SCIENTIFIC QE401, Sweden) in an ambient environment. A 4 μL droplet of deionized water was used for the measurements.

The adhesion between the current collector and the anode coating was assessed using the scratch test method. The test involved cutting parallel grooves with a 1 mm interval in perpendicular directions. The resulting grid pattern was then covered with adhesive tape (3M Scotch 610, 20 mm wide), which was smoothly peeled off.

### 2.3. Fabrication of the Si Anode

The pristine and textured Cu foils were used as current collectors. The Si anode was prepared by coating a slurry of Si nanoparticles (with a diameter of ~100 nm), PAALi, and Super P (7:2:1, wt.%) on the current collector, followed by drying at 80 °C under vacuum for 12 h. The Si mass loading ranged from 0.35 to 0.45 mg/cm^−2^.

### 2.4. Electrochemical Tests

The half-cell configurations (Coin cells, CR2032) were assembled in an argon gas-filled glovebox (Vigor SG1200/750TS, America) with a Li foil as the counter electrode and Celgard 2400 as the separator membrane. The electrolyte was 1.0 mol/L LiPF_6_ dissolved in a mixed solution of ethylene carbonate and dimethyl carbonate (1:1 vol/vol). Galvanostatic discharge–charge tests were conducted in the voltage range of 0.01–1.5 V using a battery testing instrument (LANDTE Co., China) at a current rate of 1 C (1 C = 3579 mA/g). Electrochemical impedance spectroscopy (EIS) was conducted in the frequency range of 100 kHz to 0.01 Hz at the open circuit potential with an amplitude of 5 mV using an electrochemical workstation (Bio-Logic VMP3, France).

## 3. Results and Discussion

### 3.1. Characterization of Textured Cu Foil

As illustrated in Figure 1a, the laser parameters and beam path of the laser texturing process are controlled to regulate the width (*W*), depth (*d*), and spacing (*S*) of the textured groove structure. The groove density (*D*) can be calculated using the formula $D=\frac{W}{W+S}\times 100\%$. Figure 1b shows the surface morphologies of the Cu foil in different regions. The pristine Cu foil surface exhibits a scratch-like structure at the micron scale. On the other hand, the laser-ablated area exhibits a uniform nanostructure, which is commonly observed in femtosecond laser ablation. During femtosecond laser ablation, the material experiences intense and ultrafast heating, resulting in rapid melting and vaporization. Upon cooling, a part of the vaporized material re-deposits on the surface, forming nanoparticles around the laser-ablated area [15,19]. The extremely short pulse duration of the femtosecond laser enables precise control over the energy deposition while minimizing thermal diffusion into the surrounding material. Consequently, no discernible thermal effect is observed at the boundary between the pristine and laser-ablated regions. In addition, EDS results reveal that the Cu element content on the surface of the Cu foil remains at 98.96 wt.% before and 99.21 wt.% after laser ablation, suggesting no significant oxidation. This indicates that the femtosecond laser ablation process primarily affects the surface morphology while preserving the elemental composition.

Surface structures with varying groove densities and depths are created on Cu foil with thicknesses of 4 μm, 9 μm, and 20 μm. The structure nomenclature and sizes are shown in Table 1. The thickness of the current collector in the battery plays a significant role, and opting for an ultrathin current collector is an effective way to reduce the overall weight of the battery [20]. However, achieving deeper ablation on a 4 μm-thick Cu foil is challenging. As a result of using scanning with adjacent scanning paths overlapping, an entire ablation of the Cu foil surface is achieved, resulting in the formation of only nanostructure (referred to as Cu4-nano, as shown in Figure 2a). A Cu foil with a thickness of 9 μm represents one of the most often used current collectors on the market, so it is critical to investigate its texturing and its influence on Si anode performance. Furthermore, a Cu foil with a thickness of 20 μm possesses higher mechanical strength and proves useful for investigating the enhancement mechanism of Si anode performance using deep-textured current collectors. Figure 2b,c show the typical surface morphology of the 9 μm-thick (Cu9-*D*75-*d*6) and 20 μm-thick (Cu20-*D*75-*d*15) current collectors, respectively. Both configurations exhibit a microgroove morphology, and the ablated surfaces feature nanostructures similar to those observed on the Cu4-nano current collector.

### 3.2. Electrochemical Performance of Si Anodes

The half-cell is assembled using Si nanoparticles as the anode active material. As shown in Figure 3a, the Si anode with the Cu9-P current collector experiences rapid capacity fade during the initial cycling stage and gradually depletes in subsequent cycles. This is due to the formation of the solid electrolyte interface (SEI) and the electrical isolation caused by the detachment of the expanded anode coating from the current collector, both of which are major challenges that to be addressed in Si anodes and other energy storage devices [21,22]. In contrast, Figure 3b demonstrates that the Si anode with the Cu4-nano current collector exhibits a significant increase in initial capacity from 563 mAh/g to 891 mAh/g, highlighting the positive impact of the textured nanostructure on Si anode performance. After 300 cycles at 1 C, the Cu4-nano current collector leads to a much higher discharge capacity of 463 mAh/g compared to the Si anode with the Cu9-P current collector, further proving the enhanced cycling stability provided by the textured nanostructure.

The cycling performance of the textured Cu foil with a thickness of 9 μm is presented in Figure 3c. By comparing the effects of different groove structure characteristics on Si anode performance, it is found that the initial capacity increases with larger groove densities and depths. For instance, when the groove depth *d* is 6 μm, the initial capacities of the Cu9-*D*25-*d*6, Cu9-*D*50-*d*6, and Cu9-*D*75-*d*6 anodes are 754 mAh/g, 1004 mAh/g, and 1023 mAh/g, respectively. Similarly, when the groove density *D* is 75%, the corresponding initial capacities of the Cu9-*D*75-*d*2 and Cu9-*D*75-*d*4 anodes are 761 and 994 mAh/g, respectively. The capacity retention after 300 cycles also increases with the higher groove densities and depths. The Cu9-*D*75-*d*6 anode retains the highest capacity of 911 mAh/g after 300 cycles, surpassing that of the Cu4-nano anode (463 mAh/g). This improvement is attributed to the abundant and deep microgrooves formed through laser texturing, further enhancing the cycling stability of the Si anode compared to a current collector with only surface nanostructures.

Similarly, the cycling performance of the textured Cu foil with a thickness of 20 μm is shown in Figure 3d. For a groove depth *d* of 15 μm, the initial capacity of the Cu20-*D*25-*d*15, Cu20-*D*50-*d*15, and Cu20-*D*75-*d*15 anodes are improved to 1386 mAh/g, 1307 mAh/g, and 1478 mAh/g, respectively. For a groove density *D* of 75%, the Cu20-*D*75-*d*5 and Cu20-*D*75-*d*10 anodes show initial capacities of 1229 mAh/g and 1429 mAh/g, respectively. The variation in capacity with groove density and depth follows a similar trend observed in the anodes with 9 μm- and 20 μm-thick current collectors. Among all the anodes, the Cu20-*D*75-*d*15 configuration, with the highest groove density and depth, exhibits the highest initial capacity and retained capacity after 300 cycles. The Cu20-*D*75-*d*15 anode shows an initial capacity of 1478 mAh/g with a retention of 80% (1182 mAh/g) after 300 cycles at 1 C, indicating that the more abundant and deeper ablated microgroove structure further enhances the cycling performance of the Si anode.

Furthermore, rate performance is compared to highlight the structural benefits of laser-textured current collectors. Four different current collectors, namely, the smooth Cu9-P, nanostructured Cu4-nano, and hierarchical micro/nanostructured Cu9-*D*75-*d*6 and Cu20-*D*75-*d*15, are selected for the evaluation, and the results are shown in Figure 4. The rate performance of Si anodes exhibits evident improvement when using the nanostructured Cu4-nano compared to the smooth Cu9-P. Furthermore, this improvement is further enhanced by the hierarchical micro/nanostructure and increases with higher groove density and depth. Among the evaluated anodes, the Cu20-*D*75-*d*15 configuration with the largest groove density and depth shows the highest capacity of 684 mAh/g at 3 C.

### 3.3. Discussion

Three typical fading behaviors of Si anodes are caused by the volume change of Si during the lithiation and delithiation process: (i) unstable SEI layer, (ii) pulverization of Si material, and (iii) cracks and exfoliation of the anode coating [23]. These behaviors not only result in an unstable anode structure, degrading cycling performance, but also lead to poor electric contact, reducing the rate capability. Based on the experimental results and the reasons for Si anode failure, the effect of current collector surface texturing on Si anode performance is discussed as follows.

Figure 5 illustrates the galvanostatic discharge/charge profiles of Si anodes with Cu9-P, Cu4-nano, Cu9-*D*75-*d*6, and Cu20-*D*75-*d*15 current collectors after the 2nd, 20th, and 40th cycles. In Figure 5a, it is evident that the Cu9-P anode has no obvious voltage plateau during following cycles in the galvanostatic test, and the cell voltage increases/decreases rapidly as the charging/discharging proceeds, accompanied by a rapid decay in capacity. The main reason is the expansion-induced disintegration of the anode structure, leading to significant polarization and increased internal resistance. Consequently, there is a loss of electrical contact between the Si materials and the current collector, resulting in a shortened cycling life. In Figure 5b (Cu4-nano) and 5c (Cu9-*D*75-*d*6), the anodes present a distinct voltage plateau, during which the cell voltage varies slowly and the capacity increases rapidly. The reversible and stable charging–discharging process indicates successful achievement of good electric contact, low internal resistance, and a stable structure. As shown in Figure 5d, the slope of the Cu20-*D*75-*d*15 anode’s curve is the smallest, indicating the presence of the most conductive environment within the cell. This enhanced conductivity is attributed to the nanostructures and the abundant and deep microgrooves, which provide an effective pathway for electron collection and help reduce polarization [24].

The electric contact is further validated through EIS measurement. In Figure 6, the Nyquist plots of the Si anodes with the four types of current collectors are shown at the delithiation state after the 50th cycle. The semicircle at high-medium frequency (10^3^–10^5^ Hz) mainly represents the charge transfer resistance (*R*_ct_). After laser texturing, there is a significant reduction in *R*_ct_, which continues to decrease as the groove density and depth increase. This reduction in *R*_ct_ contributes to enhanced electric contact in the Si anode. The electric contact is primarily determined by the contact area between the anode coating and the current collector, as well as the structural stability [25]. These factors are crucial in determining cycling life, as validated and analyzed below.

The roughness values of the Cu9-P, Cu4-nano, Cu9-*D*75-*d*6, and Cu20-*D*75-*d*15 current collectors are 0.275 μm, 0.569 μm, 2.052 μm, and 4.410 μm, respectively. The increased thickness of the Cu foil enables deeper microgrooves, leading to higher roughness. This larger roughness has the potential to enhance slurry wettability and facilitate the infiltration of slurry into the current collector [26]. As a result, improved electric contact is achieved, resulting in the optimal cycling and rate performance of the Si anode when using the Cu20-*D*75-*d*15 current collector with the largest roughness.

The slurry wettability is evaluated using contact angle measurement. As shown in Figure 7a, the contact angles for Cu9-P, Cu4-nano, Cu-*D*75-*d*6, and Cu-*D*75-*d*15 current collectors are 92.11°, 53.97°, 53.01°, and 46.95°, respectively. These results demonstrate that the Cu surface with nanostructures becomes more hydrophilic after laser texturing, and the hydrophilicity increases with an enlarged microgroove density and depth. The Cu-*D*75-*d*15 current collector, with the highest hydrophilicity, exhibits the most favorable slurry wettability, contributing to improved electric contact. Adhesive strength is further assessed using a scratch test. As shown in Figure 7b, the anode coating visibly peels off from the Cu9-P current collector, particularly near the scraped grid. The integrity of anode coating increases after laser texturing, improving further with increased groove density and depth. This demonstrates the improvement in adhesive strength achieved by the hierarchical nanostructures and microgrooves [17]. On the Cu20-*D*75-*d*15 current collector, the anode coating shows very little peeling.

Microscopic morphology observations are conducted to provide further insights into the aforementioned surface properties. Cross-sectional morphologies (Figure 7c) reveal that on the current collector without microgrooves, Cu4-nano presents an intact adhesive interface, whereas Cu9-P shows an obvious interface gap. This discrepancy explains the improvement in electric contact and adhesive strength, which is attributed to the laser-textured nanostructures that increase the contact area and provide a cohesive interface. In the Cu9-*D*75-*d*6 and Cu20-*D*75-*d*15 anodes, the slurry coating predominantly infiltrates the microgrooves with an integrated interface [27,28]. The microgrooves not only improve the contact area but also provide protection by creating periodic separations within the coating to prevent extrusion between adjacent expanded Si materials [29,30]. As a result, the integrity of the anode surface morphology after 50 cycles (Figure 7d), primarily due to the presence of laser-textured nanostructures (as observed in the comparison between Cu9-P and Cu4-nano), with further improvement achieved through the formation of abundant and deep microgrooves. Notably, no significant cracks or exfoliation are observed in the Si anode with the Cu20-*D*75-*d*15 current collector.

## 4. Conclusions

This study utilizes a femtosecond laser to texture the Cu foil current collector, creating a hierarchical micro/nanostructure consisting of microgroove arrays with uniform nanostructures on the surface. The effect of microgroove density and depth on the electrochemical performance of Si anodes is investigated using Cu foils with thicknesses of 4 μm, 9 μm, and 20 μm. The presence of nanostructures and microgrooves enhances the contact area between the anode coating and the current collector, leading to improved electric contact, surface wettability, and adhesive strength. Significantly, unlike previous research, the deep ablated microgroove arrays enable slurry infiltration, thereby protecting the anode coating from failure caused by volume expansion. Consequently, the cycling and rate performance of the Si anode are significantly improved, and the improvement is greater with increasing microgroove density and depth. This study presents a promising approach for improving the electrochemical performance of Si anodes through deep ablation of the current collector.

## Figures and Tables

**Figure 1 nanomaterials-13-02534-f001:**
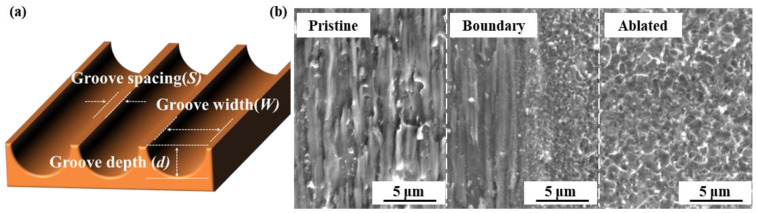
(**a**) Schematic of the laser-ablated Cu foil surface, and (**b**) surface morphologies of the Cu foil in the pristine area, ablated boundary, and ablated area.

**Figure 2 nanomaterials-13-02534-f002:**
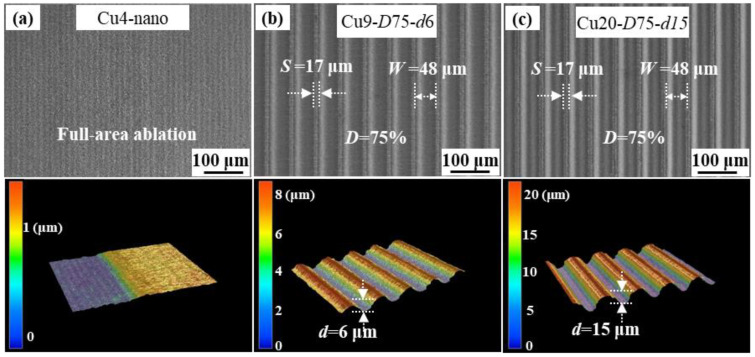
Typical surface morphology of the laser-textured Cu foil current collector: (**a**) Cu4-nano, (**b**) Cu9-*D*75-*d*6, and (**c**) Cu20-*D*75-*d*15.

**Figure 3 nanomaterials-13-02534-f003:**
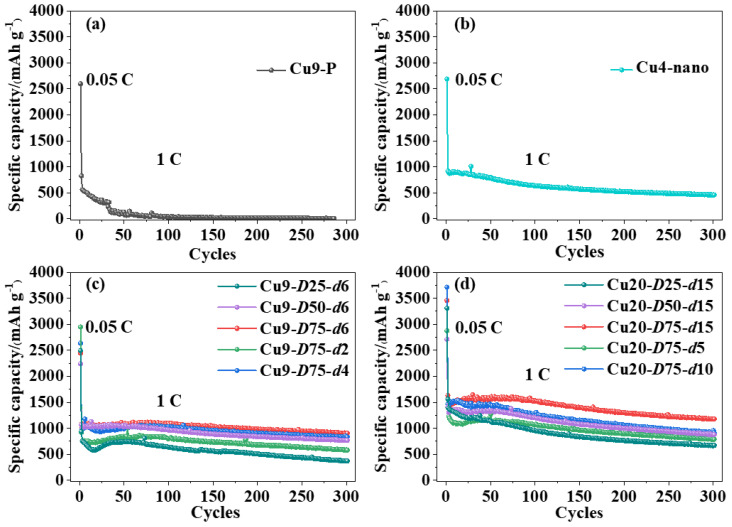
Cycling performance of the Si anodes: (**a**) Cu9-P, (**b**) Cu4-nano, (**c**) Cu9-*D*75-*d*6, and (**d**) Cu20-*D*75-*d*15.

**Figure 4 nanomaterials-13-02534-f004:**
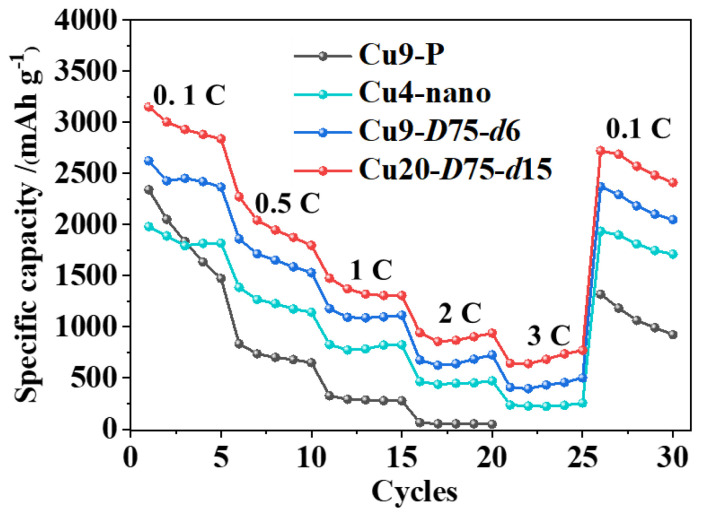
Rate performance of the Si anodes with pristine and laser-textured Cu current collectors.

**Figure 5 nanomaterials-13-02534-f005:**
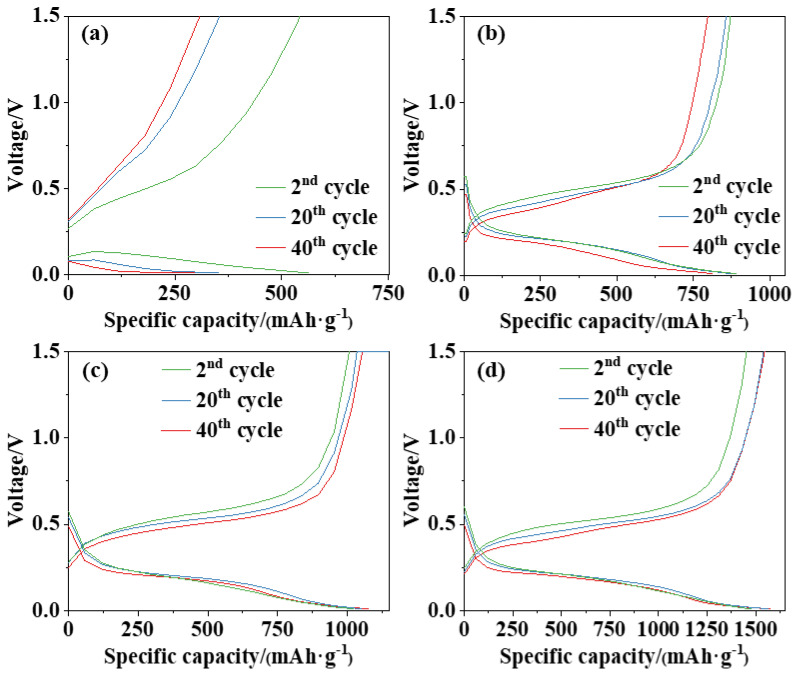
Voltage–capacity curves of the Si anodes: (**a**) Cu9-P, (**b**) Cu4-nano, (**c**) Cu9-*D*75-*d*6, and (**d**) Cu20-*D*75-*d*15.

**Figure 6 nanomaterials-13-02534-f006:**
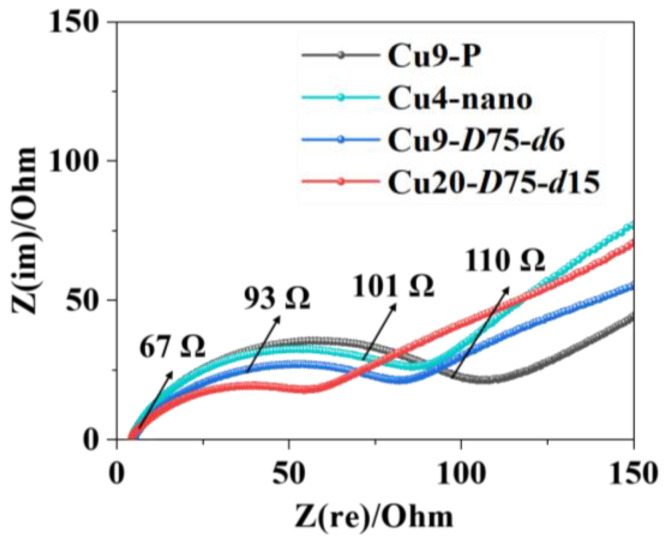
EIS results of the Si anodes with pristine and laser-textured Cu foil current collectors.

**Figure 7 nanomaterials-13-02534-f007:**
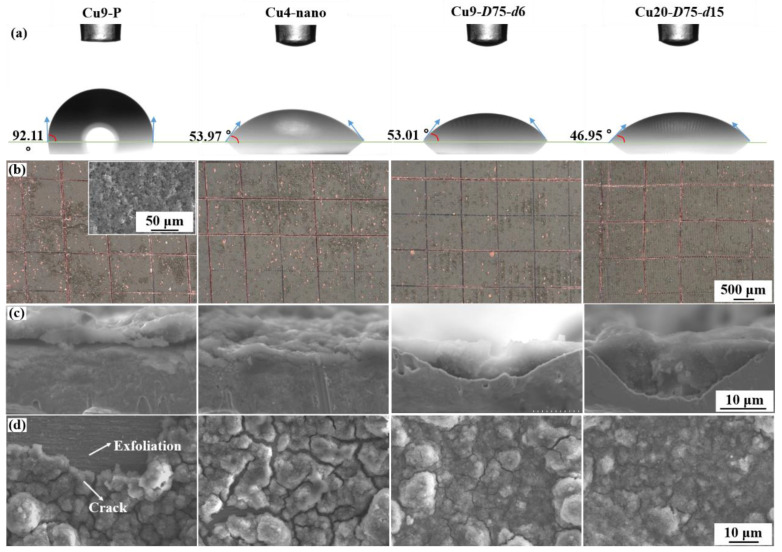
Comparison between Cu9-P, Cu4-nano, Cu9-*D*75-*d*6, and Cu20-*D*75-*d*15: (**a**) surface contact angle, (**b**) adhesion test results, and (**c**) cross-sectional and (**d**) surface morphologies of the Si anodes after 50 cycles at 1 C (note: inset in (**b**) is the typical surface morphology of Si anodes coated on the pristine and textured Cu foils).

**Table 1 nanomaterials-13-02534-t001:** Structure features of the textured Cu foil current collector.

Thickness of Cu Foil (μm)	Current Collector	*D* (%)	*d* (μm)
4	Cu4-nano	-	-
9	Cu9-P	-	-
	Cu9-*D*25-*d*6	25%	6
	Cu9-*D*50-*d*6	50%	6
	Cu9-*D*75-*d*6	75%	6
	Cu9-*D*75-*d*2	75%	2
	Cu9-*D*75-*d*4	75%	4
20	Cu20-*D*25-*d*15	25%	15
	Cu20-*D*50-*d*15	50%	15
	Cu20-*D*75-*d*15	75%	15
	Cu20-*D*75-*d*5	75%	5
	Cu20-*D*75-*d*10	75%	10

## Data Availability

Data are contained within the article.

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
