# Peer review of "Effect of Laser-Textured Cu Foil with Deep Ablation on Si Anode Performance in Li-Ion Batteries"

_nanomaterials, 2023, doi:10.3390/nano13182534_

Round 1
Reviewer 1 Report
It is required to answer the questions that have arisen and eliminate some inaccuracies
1. Throughout the presentation, it is stated that the modification of the surface of the copper foil is designed to reduce the contact resistance to the silicon electrode. However, the charge-discharge curves in Fig. 5 indicate a high resistance of the electrode, because the voltage at the beginning of the Li insertion process (0.5-0.6 V) is significantly lower than at the end of lithium extraction (1.5 V), i.e. when the current is turned on, the potential jump due to the voltage drop across the series resistance is large ~ 1 V. Accordingly, the initial voltage during the extraction of Li (~0.25 V) is significantly higher than its final value during the introduction of Li (0.01 V). No noticeable changes in these values are observed with varying the density and depth of the grooves, as well as the thickness of the initial foil from 9 to 20 μm.
2. In the bottom line of fig. Figure 2 shows a corrugated foil with both surfaces curved, while in Figure 1a the grooves are present only on the front side, while the reverse side remains flat.
3. In fig. 4 currents are not indicated.
4. In fig. 6 it is desirable to indicate the frequency at the characteristic points of the curves
5. When describing fig. 7 it is stated that in the case of an optimal configuration of the grooves, the edges of the anode coating on both sides of the scratches are smooth. This, unfortunately, is not visible.
Reviewer 2 Report
Thank you for the manuscript. Given the development stage of Si, I really think full cell results should be getting presented.
Further, the loading of these cells is very low at well below 1 mg/cm2. How do these results translate to electrodes with EV relevant loadings. The electrode contains only 70% active Si, which is quite low. The point of using Si is to make high energy density cells. However, the low loading and low fraction of active material studied here prevent that.
The paper is light on characterization. I would recommend some post-mortem analysis like SEM and XPS after cycling.
Reviewer 3 Report
I have reviewed the article entitled “Effect of laser-textured Cu foil with deep ablation on the Si anode performance in Li-ion batteries” by J. Wang submitted to Nanomaterials / MDPI. Laser ablation is a valuable and scalable technique for decreasing the effective tortuosity of Si electrodes for facile Li-ion transport deep into the electrode. Many studies were reported on various techniques to engineer electrode architectures with voids or macro-pores tailored to promote electrolyte wetting and improve Li-ion transport. However, short-pulsed lasers used to ablate micro-pores could be new.
The reported research is significant and addresses the main aspects of laser patterns and improved electrochemical performance. The manuscript itself is written OK but the main concern of this paper in terms of laser-induced electrode damage (not clearly addressed), and the areas that require clarification are given below.
· In the abstract, line 20, “stably” check the language.
· What has the laser ablation residual impact been left on the electrode material? Does it have any impact on the stoichiometry of the cathode/anode materials? Will the wavelength of the laser affect the texture?
· What is residual capacity? How much?
· Page 4, line 140, what is “lapped” scanning?
· Figure 1b, for pristine and boundary samples, SEM images must show the scale bar.
· Why there is a significant irreversible capacity after the first cycle? Although it accounts for the SEI layer and is plated on the anode the loss is significant. The irreversible loss of first discharge capacity must be promptly referred to in the recent literature (such as 10.1039/C8NR03824D; and doi.org/10.1016/j.progsolidstchem.2020.100298).
· Why do the higher grove densities and depths enhance the capacity retention?
· Do the laser ablation enhance the fast-charge capacity?
· From Figure 6, please draw the values for Rct, Rs, and other parameters.
· Why there is a significant polarization in the voltage between charge and discharge profiles?
· The number of references are too limited.
Language can be improved.
Author Response
Dear Editor,
We have carefully studied the valuable comments from you and reviewers and tried our best to revise the manuscript. The point-to-point responses to the reviewers’ comments are listed below.
Reviewer 3
I have reviewed the article entitled “Effect of laser-textured Cu foil with deep ablation on the Si anode performance in Li-ion batteries” by J. Wang submitted to Nanomaterials / MDPI. Laser ablation is a valuable and scalable technique for decreasing the effective tortuosity of Si electrodes for facile Li-ion transport deep into the electrode. Many studies were reported on various techniques to engineer electrode architectures with voids or macro-pores tailored to promote electrolyte wetting and improve Li-ion transport. However, short-pulsed lasers used to ablate micro-pores could be new.
The reported research is significant and addresses the main aspects of laser patterns and improved electrochemical performance. The manuscript itself is written OK but the main concern of this paper in terms of laser-induced electrode damage (not clearly addressed), and the areas that require clarification are given below.
- In the abstract, line 20, “stably” check the language.
Response: We acknowledge the reviewer’s comment. We have revised it in the manuscript.
<ABSTRACT>
The Si anode on the 20 μm-thick Cu current collector with a groove density of 75% and a depth of 15 μm exhibits a reversible capacity of 1181 mAh/g after 300 cycles at 1 C and shows a high-rate capacity of 719 mAh/g at 3 C.
- What has the laser ablation residual impact been left on the electrode material? Does it have any impact on the stoichiometry of the cathode/anode materials? Will the wavelength of the laser affect the texture?
Response: In the current study, EDS results reveal that there is no significant oxidation before and after laser ablation, indicating that the femtosecond laser ablation process primarily affects the surface morphology while preserving the elemental composition. Its effect is discussed in the manuscript (please see lines 127-129). The wavelength of the laser affects the texture. The shorter the laser wavelength, the lower energy density of material removal, preventing material melting caused by thermal effect and enhancing ablation quality.
- What is residual capacity? How much?
Response: Residual capacity is the capacity after 300 cycles. We have revised it to make it clearer in the revised manuscript.
<Page 1 >
The Si anode on the 20 μm-thick Cu current collector with a groove density of 75% and a depth of 15 μm exhibits a capacity of 1182 mAh/g after 300 cycles at 1 C and shows a high-rate capacity of 684 mAh/g at 3 C.
<Page 5>
The Cu9-D75-d6 anode retains the highest capacity of 911 mAh/g after 300 cycles.
<Page 6>
Among all the anodes, the Cu20-D75-d15 configuration, with the highest groove density and depth, exhibits the highest initial capacity and retained capacity after 300 cycles.
- Page 4, line 140, what is “lapped” scanning?
Response: Lapped scanning is a scan strategy with adjacent scanning paths overlapping. We have revised this sentence to make it clearer.
<Page 4>
As a result of using scanning with adjacent scanning paths overlapping, an entire ablation of the Cu foil surface is achieved, resulting in the formation of only nanostructure (referred to as Cu4-nano, as shown in Figure 2a).
- Figure 1b, for pristine and boundary samples, SEM images must show the scale bar.
Response: The scale bar has been provided.
Figure 1. (a) Schematic of the laser-ablated Cu foil surface, and (b) surface morphologies of the Cu foil in the pristine area, ablated boundary, and ablated area.
- Why there is a significant irreversible capacity after the first cycle? Although it accounts for the SEI layer and is plated on the anode the loss is significant. The irreversible loss of first discharge capacity must be promptly referred to in the recent literature (such as 10.1039/C8NR03824D; and doi.org/10.1016/j.progsolidstchem.2020.100298).
Response: We acknowledge the reviewer’s comment. The SEI formation and the electrical isolation of the fragments (formed during volume expansion) and current collector result in capacity loss. We have discussed it in the revised manuscript.
<Page 5>
This is due to the formation of the solid electrolyte interface (SEI) and the electrical isolation caused by the detachment of the expanded anode coating from the current collector, both of which are major challenges that to be addressed in Si anodes and other energy storage devices[23,24].
- Divakaran A M, Minakshi M, Bahri P A, et al. Rational design on materials for developing next generation lithium-ion secondary battery[J]. Progress in Solid State Chemistry, 2021, 62: 100298.
- Minakshi M, Mitchell D R G, Munnangi A R, et al. New insights into the electrochemistry of magnesium molybdate hierarchical architectures for high performance sodium devices[J]. Nanoscale, 2018, 10(27): 13277-13288.
- Why do the higher grove densities and depths enhance the capacity retention?
Response: The higher the grove density and depth, the more surfaces with nanostructures there are, which helps to improve the adhesion between the current collector and the coating. In addition, they provide further protection by forming more separations within the coating to prevent extrusion between adjacent expanded Si material. Both help to prevent coating exfoliation and improve capacity retention.
- Do the laser ablation enhance the fast-charge capacity?
Response: The fast-charge capacity is increased due to the laser ablation as shown in Figure 4.
- From Figure 6, please draw the values for Rct, Rs, and other parameters.
Response: We have provided the Rct value in Figure 6 in the revised manuscript.
Figure 6. EIS results of the Si anodes with pristine and laser-textured Cu foil current collectors.
- Why there is a significant polarization in the voltage between charge and discharge profiles?
Response: Because of its inherent properties, the Si anode possesses substantial polarization. Generally, improving the electrolyte, binder, conductive material, cycling stability, or employing carbon-coated Si is an effective way to increase electrode internal conductivity and reduce polarization. In this study, the Si anodes with laser processed Cu foil reduce polarization compared to the Si anode with pristine Cu foil, indicating the effect of textured current collector on polarization improvement. Our results also suggest that this improvement is limited by just texturing the current collector. To improve electrochemical performance, a number of improvement strategies are suggested to use.
- The number of references are too limited.
Response: More references have been provided.
Reference
- Cao L, Zheng M, Wang J, et al. Alloy-type lithium anode prepared by laser microcladding and dealloying for improved cycling/rate performance[J]. ACS Nano, 2022, 16(10): 17220-17228.
- Liu J, Yue M, Wang S, et al. A review of performance attenuation and mitigation strategies of lithium-ion batteries[J]. Advanced Functional Materials, 2022, 32(8): 2107769.
- Cao L, Huang T, Zhang Q, et al. Porous Si/Cu anode with high initial coulombic efficiency and volumetric capacity by comprehensive utilization of laser additive manufacturing-chemical dealloying[J]. ACS Applied Materials & Interfaces, 2020, 12(51): 57071-57078.
- Cao L, Xiao R, Wang J, et al. Recycling waste Al-Si alloy for micrometer-sized spongy si with high areal/volumetric capacity and stability in lithium-ion batteries[J]. ACS Sustainable Chemistry & Engineering, 2022, 10(25): 8143-8150.
- David R, Steeve R, Driss M, et al. An electrochemically roughened Cu current collector for Si-based electrode in Li-ion batteries[J]. Journal of Power Sources, 2013, 239: 308-314.
- Wang S E, Kim D, Kim M J, et al. Achieving cycling stability in anode of lithium-ion batteries with silicon-embedded titanium oxynitride microsphere[J]. Nanomaterials, 2023, 13(1): 132.
- Yang Y G, Zhang Z T, Yue H Y, et al. Anti-cognition in lithium-ion battery electrolytes: Comparable performance with degraded electrolyte[J]. Journal of Power Sources, 2020, 464.
- Tsao C, Yang T, Chen K, et al. Comparing the ion-conducting polymers with sulfonate and ether moieties as cathode binders for high-power lithium-ion batteries[J]. ACS Applied Materials & Interfaces, 2021, 13(8): 9846-9855.
- Zheng Y, Yin D, Seifert H J, et al. Investigation of fast-charging and degradation processes in 3d silicon-graphite anodes[J]. Nanomaterials, 2022, 12(1): 140.
- Yang Y, Yuan W, Zhang X, et al. A review on structuralized current collectors for high-performance lithium-ion battery anodes[J]. Applied Energy, 2020, 276: 115464.
- Kim S J, Moon S H, Kim M C, et al. Micro-patterned 3D Si electrodes fabricated using an imprinting process for high-performance lithium-ion batteries[J]. Journal of Applied Electrochemistry, 2018, 48: 1057-1068.
- Zhu P, Gastol D, Marshall J, et al. A review of current collectors for lithium-ion batteries[J]. Journal of Power Sources, 2021, 485: 229321.
- Kumar V, Verma R, Kango S, et al. Recent progresses and applications in laser-based surface texturing systems[J]. Materials Today Communications, 2021, 26: 101736.
- Reyter D, Rousselot S, Mazouzi D, et al. An electrochemically roughened Cu current collector for Si-based electrode in Li-ion batteries[J]. Journal of Power Sources, 2013, 239: 308-314.
- Petare A C, Mishra A, Palani I A, et al. Study of laser texturing assisted abrasive flow finishing for enhancing surface quality and microgeometry of spur gears[J]. The International Journal of Advanced Manufacturing Technology, 2019, 101: 785-799.
- Allahyari E, JJ Nivas J, Oscurato S L, et al. Laser surface texturing of copper and variation of the wetting response with the laser pulse fluence[J]. Applied Surface Science, 2019, 470: 817-824.
- Wang Y, Zhao Z, Zhong J, et al. Hierarchically micro/nanostructured current collectors induced by ultrafast femtosecond laser strategy for high-performance lithium-ion batteries[J]. Energy & Environmental Materials, 2022, 5(3): 969-976.
- Zhang N, Zheng Y, Trifonova A, et al. Laser structured Cu foil for high-performance lithium-ion battery anodes[J]. Journal of Applied Electrochemistry, 2017, 47(7): 829-837.
- Li Q, Sun X, Zhao W, et al. Processing of a large-scale microporous group on copper foil current collectors for lithium batteries using femtosecond laser[J]. Advanced Engineering Materials, 2020, 22(12): 2000710.
- Anoop K K, Fittipaldi R, Rubano A, et al. Direct femtosecond laser ablation of copper with an optical vortex beam[J]. Journal of Applied Physics, 2014, 116(11): 113102.
- Allahyari E, Nivas J J J, Oscurato S L, et al. Laser surface texturing of copper and variation of the wetting response with the laser pulse fluence[J]. Applied Surface Science, 2019, 470: 817-824.
- Choudhury R, Wild J, Yang Y. Engineering current collectors for batteries with high specific energy[J]. Joule, 2021, 5(6): 1301-1305.
- Divakaran A M, Minakshi M, Bahri P A, et al. Rational design on materials for developing next generation lithium-ion secondary battery[J]. Progress in Solid State Chemistry, 2021, 62: 100298.
- Minakshi M, Mitchell D R G, Munnangi A R, et al. New insights into the electrochemistry of magnesium molybdate hierarchical architectures for high performance sodium devices[J]. Nanoscale, 2018, 10(27): 13277-13288.
- Kwon T, Choi J W, Coskun A. The emerging era of supramolecular polymeric binders in silicon anodes[J]. Chemical Society Reviews, 2018, 47(6): 2145-2164.
- Li S, He Q, Chen K, et al. Facile chemical fabrication of a three-dimensional copper current collector for stable lithium metal anodes[J]. Journal of The Electrochemical Society, 2021, 168(7): 070502.
- Cao F F, Deng J W, Xin S, et al. Cu-Si nanocable arrays as high-rate anode materials for lithium-ion batteries[J]. Advanced materials, 2011, 23(38): 4415-4420.
- Baldan A. Adhesion phenomena in bonded joints[J]. International Journal of Adhesion and Adhesives, 2012, 38: 95-116.
- Pfleging W. Recent progress in laser texturing of battery materials: a review of tuning electrochemical performances, related material development, and prospects for large-scale manufacturing[J]. International Journal of Extreme Manufacturing, 2021, 3(1).
- Jeon H, Cho I, Jo H, et al. Highly rough copper current collector: improving adhesion property between a silicon electrode and current collector for flexible lithium-ion batteries[J]. RSC Advances, 2017, 7(57): 35681-35686.
- Pfleging W. A review of laser electrode processing for development and manufacturing of lithium-ion batteries[J]. Nanophotonics, 2018, 7(3): 549-573.
- So J Y, Moon S H, Kim M C, et al. Stress dispersed cu metal anode by laser multiscale patterning for lithium-ion batteries with high capacity[J]. Metals, 2018, 8(6): 410.

Reviewer 4 Report
This manuscript provides a method to overcome the typical peeling and cracking phenomenon in the Si anode by adopting Cu foils with hierarchical structure. The deep-ablated Cu foil reduced interfacial resistance to promote electron transfer and thus improved rate capability. The authors systematically studied the role of femtosecond laser to make textured Cu foil and demonstrated the improved electrochemical properties. Therefore, this manuscript can be published in this journal after dealing with the following minor comments and questions.
(1) During the laser ablation, the material experiences rapid melting and vaporization and then the vaporized Cu components re-condensed on the surface. Is there any weight change before and after the laser ablation?
(2) The surface morphologies of Si anodes coated on laser treated Cu foils should be provided.
(3) Along with the important role of Cu foils that has hierarchical structure, the loading density seems important to fully understand the effect of the laser ablation. According to the information written in Experimental part, the Si mass loading is ranged from 0.35 to 0.45 mg/cm2. Can authors further increase the loading density and provide the electrochemical properties?
(4) It seems the laser-texture of Cu foil is one directional. Based on Author’s finding, the laser treatment in the form of crosses by processing both vertically and horizontally is expected to provide more grooves that can possess more Si particle. Can authors provide opinion about this?
Author Response
Reviewer 4
This manuscript provides a method to overcome the typical peeling and cracking phenomenon in the Si anode by adopting Cu foils with hierarchical structure. The deep-ablated Cu foil reduced interfacial resistance to promote electron transfer and thus improved rate capability. The authors systematically studied the role of femtosecond laser to make textured Cu foil and demonstrated the improved electrochemical properties. Therefore, this manuscript can be published in this journal after dealing with the following minor comments and questions.
1.During the laser ablation, the material experiences rapid melting and vaporization and then the vaporized Cu components re-condensed on the surface. Is there any weight change before and after the laser ablation?
Response: As only a part of vaporized Cu components re-condenses on the surface, the weight of Cu foil decreases after the laser ablation. For example, the weight of 20 μm thick Cu foil before and after laser ablation (Cu20-D75-d15) is about 17.6 mg and 12.3 mg, respectively.
2.The surface morphologies of Si anodes coated on laser treated Cu foils should be provided.
Response: We have provided the typical surface morphology in the revised manuscript.
<page 10>
Figure 7. Comparison between Cu9-P, Cu4-nano, Cu9-D75-d6, and Cu20-D75-d15: (a) surface contact angle, (b) adhesion test results, (c) cross-sectional and (d) surface morphologies of the Si anodes after 50 cycles at 1 C. (Note: inset in (b) is the typical surface morphology of Si anodes coated on the pristine and textured Cu foils)
- Along with the important role of Cu foils that has hierarchical structure, the loading density seems important to fully understand the effect of the laser ablation. According to the information written in Experimental part, the Si mass loading is ranged from 0.35 to 0.45 mg/cm2. Can authors further increase the loading density and provide the electrochemical properties?
Response: We acknowledge the reviewer’s comment. Increasing the loading leads to rapid degradation of electrode performance, as shown in the following figure, which shows the cyclic performance with a loading of 0.544 mg/cm2. The degradation with increasing loading is also found in previous studies, which is influenced by several factors, primarily binder strength, coating uniformity, and so on. In most of the current Si anode studies, the loading remains below 1 mg/cm2. This study specifically focuses on examining the interfacial influence between the electrode coating and the current collector. Consequently, we employed a smaller loading and a higher binder ratio to achieve a more stable electrode, thereby minimizing the influence of other factors. The interfacial influence obtained in this study serves as a basis for understanding high-loading electrodes, despite the involvement of more complex factors. We will further increase the loading density in future work.
- It seems the laser-texture of Cu foil is one directional. Based on Author’s finding, the laser treatment in the form of crosses by processing both vertically and horizontally is expected to provide more grooves that can possess more Si particle. Can authors provide opinion about this?
Response: Laser processing can create more grooves by scanning in both vertical and horizontal directions, posing more Si particles. However, the electrochemical properties are determined by factors other than Si concentration. For example, increased Cu removal results in a loss of certain strength of Cu foil (see ref. 10.1021/acsami.1c13233); the smaller size and higher density of the columnar structure created by laser cross-scanning lower the volume expansion protection effect.

Round 2
Reviewer 2 Report
I don't feel the authors have addressed my comments/concerns enough.
English is ok.
Author Response
We have tried our best to address the reviewer’s comments.
Reviewer 3 Report
The revised version is satisfactory.